# IMPROVED UNCERTAINTY POST-CALIBRATION VIA RANK PRESERVING TRANSFORMS

## ABSTRACT

Modern machine learning models with high accuracy often exhibit poor uncertainty calibration: the output probabilities of the model do not reflect its accuracy, and tend to be over-confident. Existing post-calibration methods such as temperature scaling recalibrate a trained model using rather simple calibrators with one or few parameters, which can have a rather limited capacity. In this paper, we propose Neural Rank Preserving Transforms (NRPT), a new post-calibration method that adjusts the output probabilities of a trained classifier using a calibrator of higher capacity, while maintaining its prediction accuracy. NRPT learns a calibrator that preserves the rank of the probabilities through general monotonic transforms, individualizes to the original input, and allows learning with any loss function that encourages calibration. We show experimentally that NRPT improves the expected calibration error (ECE) significantly over existing post-calibration methods such as (local) temperature scaling on large-scale image and text classification tasks. The performance of NRPT can further match ensemble methods such as deep ensembles, while being much more parameter-efficient. We further demonstrate the improved calibration ability of NRPT beyond the ECE metric, such as accuracy among top-confidence predictions, as well as optimizing the tradeoff between calibration and sharpness.

## 1 INTRODUCTION

Modern machine learning models such as deep neural networks have achieved high performance on many challenging tasks, and have been put into production that impacts billions of people (LeCun et al., 2015). It is increasingly critical that the outputs of these models are comprehensible and safe to use in downstream applications. However, high-accuracy classification models often exhibit the failure mode of *miscalibration*: the output probabilities of these models do not reflect the true accuracies, and tend to be over-confident (Guo et al., 2017; Lakshminarayanan et al., 2017). As the output probabilities are typically comprehended as (an estimate of) true accuracies and used in downstream applications, miscalibration can negatively impact the decision making, and is especially dangerous in risk-sensitive domains such as medical AI (Begoli et al., 2019; Jiang et al., 2012) or self-driving cars (Michelmore et al., 2018). It is an important question how to properly calibrate these models so as to make the output probabilities more trustworthy and safer to use.

Existing methods for uncertainty calibration can roughly be divided into two types. *Diversity-based methods* such as ensembles (Lakshminarayanan et al., 2017; Wen et al., 2020) and Bayesian networks (Gal & Ghahramani, 2016; Maddox et al., 2019; Dusenberry et al., 2020) work by aggregating predicted probability over multiple models or multiple times on a randomized model. These methods are able to improve both the accuracy and the uncertainty calibration over a single deterministic model (Ovadia et al., 2019). However, deploying these models requires either storing all the ensemble members and/or running multiple random variants of the same model, which makes them memory-expensive and runtime-inefficient. On the other hand, *post-calibration methods* work by learning a *calibrator* on top of the output probabilities (or logits) of an existing well-trained model (Platt et al., 1999; Zadrozny & Elkan, 2001; 2002; Guo et al., 2017; Ding et al., 2020). For a $K$-class classification model that outputs logits $z = z(x) \in \mathbb{R}^K$, post-calibration methods learn a calibrator $f : \mathbb{R}^K \to \mathbb{R}^K$ using additional holdout data, so that $f(z)$ is better calibrated than the original $z$. The architectures of such calibrators are typically simple: A prevalent example is the temperature scaling method which learns $f_T(z) = z/T$ with a single trainable parameter $T > 0$ by

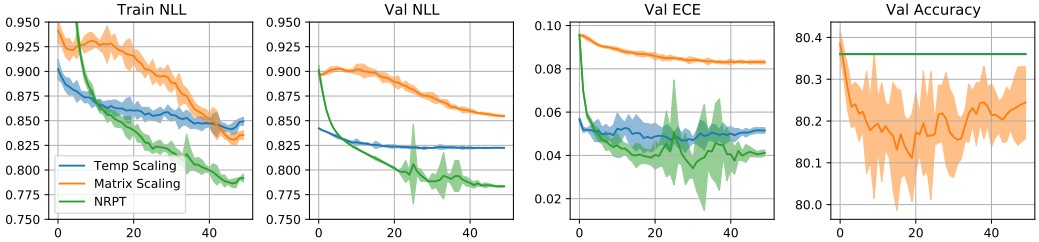

Figure 1: Post-calibration training curves on a WideResNet-28-10 on CIFAR-100. **Temperature scaling** minimizes the training and validation NLL reasonably well (and improves the ECE), but still underfits the NLL. **Matrix scaling** learns a higher-capacity matrix calibrator and minimizes the training NLL better, but does not improve the ECE since the calibrator does not maintain the accuracy and is encouraged to improve the accuracy instead of calibration. Our **Neural Rank-Preserving Transforms (NRPT)** learns a higher-capacity calibrator that **preserves the accuracy**, and improves both the training/validation NLL as well as the ECE.

minimizing the negative log-likelihood (NLL) loss on holdout data. Such simple calibrators add no overhead to the existing model, and is empirically shown to improve the calibration significantly on a variety of tasks and models (Guo et al., 2017).

Despite its empirical success, the design of post-calibration methods is not yet fully satisfactory: In practice, simple calibrators such as temperature scaling often underfit the calibration loss on its training data, whereas more complex calibrators can often overfit—see Figure 1 for a quantitative illustration of this effect. While the underfitting of simple calibrators are perhaps due to their limited expressive power, the overfitting of complex calibrators is also believed to be natural since the holdout dataset used for training the calibrators are typically small (e.g. a few thousands of examples). One concrete example is the *matrix scaling* method which learns a matrix calibrator $f_{W,b}(z) = Wz + b$ involving $O(K^2)$ trainable parameters. When $K$ is large, matrix scaling often tend to overfit and hurt calibration, despite being a strict generalization of temperature scaling (Guo et al., 2017). It is further observed that the overfitting cannot be easily fixed by applying common regularizations such as $L_2$ on the calibrator (Kull et al., 2019). These empirical evidence seems to suggest that complex calibrators with a large amount of parameters are perhaps not recommended in designing post calibration methods.

In this paper, we show that in contrast to the prior belief, large calibrators do not necessarily overfit; it is rather a *lack of accuracy constraint* on the calibrator that may have caused the overfitting. Observe that matrix scaling, unlike temperature scaling, *is not guaranteed to maintain the accuracy* of the model: it applies a general affine transform $z \mapsto Wz + b$ on the logits and can modify their rank (and thus the predicted top label), whereas temperature scaling is guaranteed to preserve the rank. When trained with the NLL loss, a calibrator that does not maintain the accuracy may attempt to improve the accuracy at the cost of hurting (or not improving) the calibration.

Motivated by this observation, this paper proposes Neural Rank-Preserving Transforms (NRPT), a method for learning calibrators that maintain the accuracy of the model, yet are complex enough for yielding better calibration performance than simple calibrators such as temperature scaling. Our key idea is that a sufficient condition for the calibrator to maintain the accuracy is for it to *preserve the rank* of the logits: any mapping that preserves the rank of the logits will not change the predicted top label. We instantiate this idea by designing a family of calibrators that perform entrywise monotone transforms on each individual logit (or log-probability): for a $K$-class classification problem, NRPT scales each logit as $z_i \to f(z_i, x)$, where $z_i \in \mathbb{R}$ is the $i$-th logit ($1 \le i \le K$), $x \in \mathbb{R}^d$ is the original input features, and $f : \mathbb{R} \times \mathbb{R}^d \to \mathbb{R}$ is monotonically increasing in its first argument but otherwise arbitrary. As $f$ is monotone, we have $f(z_1, x) \le f(z_2, x)$ if $z_1 \le z_2$, and thus $f$ preserves the rank of the logits. This method strictly generalizes temperature scaling (which uses $f(z_i, x) = z_i/T$) and local temperature scaling (which uses $f(z_i, x) = z_i/T(x)$) (Ding et al., 2020). The fact that $f$ can depend on $x$ further helps improve the expressivity of $f$ and allows great flexibility in the architecture design. We compare our instantiation of NRPT against temperature scaling and matrix scaling in Figure 1, in which we see that NRPT is indeed able to fit the training loss better than temperature scaling and does not suffer from overfitting.

**Our contributions** We propose Neural Rank-Preserving Transforms (NRPT), an improved method for performing uncertainty post-calibration on a trained classifier while maintaining its accuracy (Section 3). NRPT learns calibrators that scale the logits using general monotone transforms, are individualized to the original input features, and allow learning with any calibration loss function (not restricted to those that correlates with the accuracy). We show experimentally that NRPT improves the expected calibration error (ECE) significantly over existing post-calibration methods on large-scale image and text classification tasks such as CIFAR-100, ImageNet, and MNLI (Section 4.1). NRPT can further match diversity-based methods such as deep ensembles, while using a much less number of additional parameters. We further demonstrate the strong calibration ability of NRPT beyond the ECE, by showing that it improves on the accuracy among top-confidence predictions, as well as the tradeoff between ECE and sharpness of prediction (Section 4.2).

Due to the space constraint, we defer the discussions of additional related work to Appendix A and additional experimental details and results to the later Appendices.

## 2 BACKGROUND ON UNCERTAINTY CALIBRATION

We consider $K$-class classification problems where $X \in \mathbb{R}^d$ is the input (features), $Y \in [K] := \{1, \ldots, K\}$ is the true label, and $(X, Y)$ follows some underlying joint distribution. Let $\widehat{p} : \mathbb{R}^d \to \Delta_K$ be a prediction model (for example, a neural network learned from data) that maps inputs to probabilities, where $\Delta_K := \{(p_1, \ldots, p_K) : p_i \geq 0, \sum_i p_i = 1\}$ is the set of all probability distributions on $[K]$. We say $\widehat{p}$ is *perfectly calibrated* if

$$\mathbb{P}(Y = k \mid \widehat{p}(X) = p) = p_k \quad \text{for all } p \in \Delta_K, \ k \in [K].$$

In other words, a model is perfectly calibrated if when the model predicts $\widehat{p}(X) = p$, the conditional distribution of $Y$ is exactly $p$. It is difficult to evaluate perfect calibration from finite data, as for almost all $p$ we do not receive samples that satisfy the exact conditioning $\widehat{p}(x) = p$. This motivates considering alternative scalar metrics for calibration that can be estimated from data.

**ECE** The Expected Calibration Error (ECE) is a commonly used metric that measures calibration through grouping examples according to the confidence (i.e. the top predicted probability) (Naeini et al., 2015; Guo et al., 2017). Let $\{(x_i, y_i)\}_{i=1}^n$ be the evaluation dataset on which we wish to evaluate the calibration of a model $\widehat{p}$. Define the intervals $I_m = (\frac{m-1}{M}, \frac{m}{M}]$, where $M > 0$ is a (fixed) number of bins, and partitions the examples into $M$ bins according to the confidence: $B_m = \{i : \max_k \widehat{p}(x_i)_k \in I_m\}$. Define the *accuracy* and *confidence* within $B_m$ as

$$\text{acc}(B_m) := \frac{1}{B_m} \sum_{i \in B_m} \mathbf{1}\left\{ \arg\max_k \widehat{p}(x_i)_k = y_i \right\} \quad \text{and} \quad \text{conf}(B_m) := \frac{1}{B_m} \sum_{i \in B_m} \max_k \widehat{p}(x_i)_k$$

The ECE is then defined as the (weighted) average difference between accuracy and confidence:

$$\text{ECE}(\widehat{p}) := \sum_{m=1}^{M} \frac{|B_m|}{n} |\text{acc}(B_m) - \text{conf}(B_m)|. \tag{1}$$

The ECE is a sensible calibration metric since it is a binned approximation of the *top-label calibration error* (TCE) that measures the difference between accuracy and confidence under exact conditioning:

$$\text{TCE}(\widehat{p}) := \mathbb{E}\left[\left| \mathbb{P}\left( \arg\max_k \widehat{p}(X)_k = Y \mid \max_k \widehat{p}(X)_k \right) - \max_k \widehat{p}(X)_k \right|\right].$$

**Debiased ECE** Recent work shows that the ECE has an inherent positive bias and proposes the *Debiased ECE* that approximately removes this bias using Gaussian bootstrapping (Kumar et al., 2019):

$$\text{DECE}(\widehat{p}) := \text{ECE}(\widehat{p}) - \left( \mathbb{E}_{R_{1:M}}\left[ \sum_{m=1}^{M} \frac{|B_m|}{n} |\text{conf}(B_m) - R_m| \right] - \text{ECE}(\widehat{p}) \right),$$

where $R_m \sim \mathsf{N}(\text{acc}(B_m), \frac{\text{acc}(B_m)(1 - \text{acc}(B_m))}{|B_m|})$. Kumar et al. (2019) showed that the debiased ECE is a typically more accurate estimator of the TCE than ECE, especially when the TCE is relatively small. In our experiments, we use both the ECE and the debiased ECE for evaluating calibration.

**NLL** The Negative Log-Likelihood (NLL), typically used as loss function for training classifiers, is also a measure of calibration:

$$\text{NLL}(\widehat{p}) := \frac{1}{n} \sum_{i=1}^{n} -\log \widehat{p}(x_i)_{y_i}. \tag{2}$$

NLL is a proper scoring rule (Lakshminarayanan et al., 2017) in the sense that the population minimizer over all possible $\widehat{p}$ is achieved at the ground truth conditional distribution $p_\star$ (Hastie et al., 2009). In general, the NLL measures the distance between $\widehat{p}$ and $p_\star$, and is thus a joint metric of accuracy and calibration.

**Predictive entropy (sharpness)** While we are mostly concered about the accuracy and calibration of a model, these two metrics alone do not fully guarantee a proper uncertainty quantification. For example, any high-accuracy model can be calibrated in a "trivial" way such that the ECE becomes exactly 0, by mapping the confidence on all examples to be equal to the (overall) accuracy of the model, and rescaling the non-top probabilities accordingly. In order to prevent such trivial calibration, we additionally measure the *sharpness* of the predictions using the predictive entropy (Lakshminarayanan et al., 2017):

$$\text{PEnt}(\hat{p}) = \frac{1}{n} \sum_{i=1}^{n} \sum_{k=1}^{K} -\widehat{p}(x_i)_k \log \widehat{p}(x_i)_k.$$

Lower predictive entropies indicate sharper predictions (i.e. predictions closer to delta distributions than the uniform distribution). In general, the predictive entropy is not necessarily related to the calibration; however, for models that have the same accuracies, we observe that the predictive entropy is typically negatively correlated with calibration—sharper predictions are usually less calibrated.

## 3 RANK PRESERVING TRANSFORMS

We now introduce our main algorithm Neural Rank-Preserving Transforms (NRPT) for performing post-calibration on trained classifiers. Throughout this section, we consider $K$-class classification problems ($K \geq 2$), and let $\widehat{z} : \mathbb{R}^d \to \mathbb{R}^K$ denote the input-to-logit mapping of a trained classifier. The predicted probabilities of the model is the softmax of the logits: $\widehat{p}(x) = \sigma_{\text{SM}}(\widehat{z}(x)) = [\exp(\widehat{z}(x)_k)/\sum_{j\in[K]} \exp(\widehat{z}(x)_j)]_k$.

**Temperature Scaling** We begin by reviewing temperature scaling, a simple yet strong baseline method for post-calibration. Temperature scaling recalibrates a model by scaling down the logits using a single *temperature* parameter $T > 0$:

$$f_T(\widehat{z}) = f_T(\widehat{z}_1, \ldots, \widehat{z}_K) = [\widehat{z}_1/T, \ldots, \widehat{z}_K/T] = \widehat{z}/T, \tag{3}$$

and using $\sigma_{\text{SM}}(f_T(\widehat{z}))$ as calibrated probabilities. The parameter $T$ is typically learned by minimizing the NLL loss on a hold-out calibration dataset. Temperature scaling clearly preserves the rank of the logits, and is observed to improve both the NLL and the ECE on test data by learning a temperature parameter that is typically above one (Guo et al., 2017) on large, over-confident models. However, as we have seen in Figure 1, temperature scaling often does not minimize the (training) NLL on the calibration dataset well due to its limited model capacity.

**Individualization** The first building block of our algorithm is to *individualize* (or localize) temperature scaling, an idea recently proposed in the Local Temperature Scaling (LTS) method Ding et al. (2020): Each input $x \in \mathbb{R}^d$ can have its own temperature $T(x) > 0$. This still preserves the rank of the logits, but can substantially increase the capacity of the calibrator as now the temperature can adapt to the raw input. Formally, we calibrate the model by scaling down the logits using an individualized temperature model $T_\theta(x) > 0$:

$$f_{T_\theta}(\widehat{z}; x) = [\widehat{z}_1/T_\theta(x), \ldots, \widehat{z}_K/T_\theta(x)]. \tag{4}$$

We require the temperature model $T_\theta(x)$ to always output a positive scalar, but can otherwise have arbitrary architectures. In our experiments, we find LTS with the right choice of architecture can consistently outperform temperature scaling.

**Rank-preserving transforms via general monotone calibrators**   We now introduce our key idea of performing *general monotone calibration*, which preserves the rank of the logits and can be even more flexible than local temperature scaling. Our key observation is that the fundamental property that allows (local) temperature scaling (3) and (4) to preserve the rank is their monotone property:

$$\widehat{z}_i \leq \widehat{z}_j \quad \text{guarantees} \quad f(\widehat{z})_i \leq f(\widehat{z})_j.$$

Further, this is satisfied for (3) and (4) since the calibrator applies an *entrywise, monotonically increasing function* to each logit. Motivated by this, we consider general monotone calibrators of the form

$$f_\theta(\widehat{z}; x) = [g_\theta(\widehat{z}_1; x), \ldots, g_\theta(\widehat{z}_K; x)], \quad g(z, x) \text{ is monotonically increasing in } z \text{ for all } x. \quad (5)$$

Observe that both temperature scaling ($g(z_i; x) = z_i/T$) and local temperature scaling ($g(z_i; x) = z_i/T(x)$) as special cases of (5), and under this perspective are still limited in capacity as the $g$ used in both cases are *linear* in $z_i$ given any $x$. To let the calibrator be more expressive, we would rather like to learn an arbitrary $g$ under the monotonicity constraint.

**Instantiation via monotone two-layer networks**   We now explain how we design a function class $g_\theta(z; x)$ that is monotone in $z$ for any $x$ and not too restricted in its expressivity. Existing techniques for building such monotone function classes include either classical non-parametric methods such as isotonic regression (Barlow & Brunk, 1972), or sophisticated tricks such as parametrizing the derivative $\frac{d}{dz}g_\theta(z; x)$ using a non-negative neural network (Wehenkel & Louppe, 2019). However, for the purpose designing calibrators, we prefer a simpler parametric class that enables efficient gradient-based learning. We acheive this by using a class of two-layer neural networks in $z_i$ with coefficients depending on $x$:

$$g_\theta(z_i; x) := \sum_{j=1}^{M} a_j \phi\left(\frac{1}{T_{\theta_j}(x)}\left(z_i - b_{\theta_j}(x)\right)\right), \quad (6)$$

$$\text{where } a_j \geq 0, \ T_{\theta_j}(x) > 0, \text{ and } \phi \text{ is a monotonically increasing nonlinearity.}$$

It is straightforward to see that (6) is guaranteed to be monotonically increasing in $z_i$ for any $x$ as desired. Further, by choosing a proper $\phi$ and using a large number of neurons $M$, (6) can express a fairly large class of monotonic functions in $z_i$ for any fixed $x$. We also note that (6) recovers local temperature scaling (4) if we take $\phi(t) = t$ to be the identity mapping (as $g$ becomes linear in $z_i$ in that case), and therefore has a strictly higher expressivity.

**Architectural choices**   For implementing the calibrator (5) and (6), in theory one is free to use any architecture as long as $T_{\theta_j}(x) > 0$ is guaranteed. However, we observe experimentally that *reusing the representation of the trained classifier* and *weight sharing* can help improve the calibration performance. In all our experiments, we choose $[T_{\theta_j}(x), b_{\theta_j}(x)]$ to be a two-layer neural network of the last hidden layer (the pre-logit layer) of the trained classifier, with shared weights:

$$[T_{\theta_1}(x), \ldots, T_{\theta_M}(x)] = \sigma_{\text{temp}}\Big(A_{\text{temp}}\sigma(W\widehat{H}(x) + b)\Big),$$
$$[b_{\theta_1}(x), \ldots, b_{\theta_M}(x)] = A_{\text{bias}}\sigma(W\widehat{H}(x) + b), \quad (7)$$

where $\widehat{H} : \mathbb{R}^d \to \mathbb{R}^{d_{\text{hid}}}$ is the last hidden layer of the trained classifier, and $W \in \mathbb{R}^{N \times d_{\text{hid}}}$, $b \in \mathbb{R}^N$, and $A_{\text{temp}}, A_{\text{bias}} \in \mathbb{R}^{M \times N}$ are the trainable parameters. We further use a strictly positive nonlinearity $\sigma_{\text{temp}}$ to guarantee the temperatures are positive and not too small.

**Flexible loss functions**   Our final observation is that our calibrator (5) and (6) can be trained with not only the NLL loss, but any other loss function that encourages calibration. This is only possible for rank-preserving calibrators—calibrators that do not preserve the accuracy have to be trained using a loss that jointly encourages high accuracy and calibration (such as the NLL), otherwise the calibrator may hurt the accuracy for the sake of getting good calibration.

We specifically propose to use the ECE (1) directly as a loss function for training the calibrator. Notice that even though the ECE is non-smooth in the model outputs (due to the binning and the non-smoothness of the argmax prediction rule), it still has a non-trivial gradient since the average

confidence $\mathrm{conf}(B_m)$ is differentiable with respect to the model outputs. We find that training with the ECE loss can often result in better calibration, at the cost of reducing the sharpness (see Section 4) for details). We remark that Kumar et al. (2018) has considered training the MMCE (maximum mean calibration error), a kernelized version of the ECE loss; however, we are not aware of prior work that has considered training the ECE directly to the best of our knowledge.

**Summary of algorithm**   We summarize our NRPT algorithm as follows: Build a calibrator $f_\theta(z; x)$ using the rank-preserving transform $g_\theta(z_i; x)$ defined in (5), (6), (7); Train the calibrator $f_\theta$ via minimizing any desired loss (e.g. NLL or ECE) on a holdout calibration dataset; Output the calibrated model $\widehat{p}(x) = \sigma_{\mathrm{SM}}(f_\theta(\widehat{z}(x), x))$.

# 4    EXPERIMENTS

## 4.1    CALIBRATION ON IMAGE AND TEXT CLASSIFICATION

**Tasks and models**   We perform calibration experiments on the three benchmark tasks:

- **CIFAR-100**, splitted into 45K/5K/10K as train/calibration/val data. We train a WideResNet-28-10 (Zagoruyko & Komodakis, 2016) on the train split, achieving $80.36\%$ accuracy.

- **ImageNet** ILSVRC2012 (Deng et al., 2009), splitted into 1.1M/100K/50K as train/calibration/val data. We train a WideResNet-50-2 on the train split, achieving $76.27\%$ top-1 acccuracy.

- **MNLI**, one of the largest text classification tasks in the GLUE benchmark (Wang et al., 2018), splitted into 350K/43K/20K as train/calibration/val data. We finetune a pretrained BERT-Base (Devlin et al., 2018) model, achieving $83.36\%$ accuracy on the matched (MNLI-m) data and $83.77\%$ accuracy on the mismatched (MNLI-mm) data. Calibration performances are also evaluated on MNLI-m and MNLI-mm separately.

**Methods and evaluation metrics**   We implement our NRPT algorithm with a two-layer neural network calibrator on top of the last hidden representation $\widehat{H}$ of the trained classifier (see (6) and (7).) In the case of BERT, $\widehat{H}$ is the last encoder layer at the CLS token. For all three tasks, we choose hidden dimension $N = 512$, the number or neurons $M \in \{5, 10\}$, and $\phi$ to be the leaky-relu activation. We choose $\sigma_{\mathrm{temp}}(t) = 0.2 + \mathrm{relu}_6(t)$, so that $T_{\theta_i}(x)$ is guaranteed to be within $[0.2, 6.2]$.

We compare our NRPT against the original uncalibrated model, as well as two existing post-calibration methods: **Temperature Scaling** (TS, see (3)), a strong baseline method for post-calibration (Guo et al., 2017), and **Local Temperature Scaling** (LTS, see (4)), a generalization of temperature scaling that is observed to achieve state-of-the-art performance on a variety of computer vision tasks (Ding et al., 2020)[1]. On CIFAR-100 and ImageNet we further compare with deep ensembles (Lakshminarayanan et al., 2017), a representative diversity-based method with strong calibration performance (Ovadia et al., 2019). We train the calibrators using either the NLL loss (2) or the ECE loss (1) with *unregularized* minibatch SGD on the calibration data.[2] We use "+E" in {TS+E, LTS+E, NRPT+E} to indicate that a method is trained with the ECE loss. Additional architectural and training details can be found in Appendix B.

We evaluate the calibration methods on three metrics: the NLL, the ECE and debiased ECE (DECE), as well as the predictive entropy (PEnt) for evaluating the sharpness of the calibrated predictions.

**Results**   Table 1 summarizes our main results. We observe that NRPT and NRPT+E consistently performs the best among each group in terms of both the NLL and the ECE metric. In particular, the best test likelihood on CIFAR100 and ImageNet are achieved by NRPT and the best test ECE or debiased ECE are achieved by either NRPT or NRPT+E. This justifies our intuition that maximizing the expressivity of the calibrator while making sure it preserves the rank of the logits can indeed

---

[1]We remark that our method is partly motivated by the attempt to improve over matrix scaling. However, we find matrix scaling consistently performs worse than temperature scaling, and thus we omit the results here.

[2]For temperature scaling, since there is only one trainable parameter, we can in theory obtain the exact optimal solution on the entire dataset; however we observed that the SGD solution with proper learning rate decay almost always coincides with the exact solution.

Table 1: Comparison between NRPT and existing post-calibration methods. Metrics are reported in terms of the mean and standard deviation over 4 random seeds.

| Task | Metric | Uncal | TS | LTS | **NRPT** | TS+E | LTS+E | **NRPT+E** |
|---|---|---|---|---|---|---|---|---|
| CIFAR100 | NLL | 0.90 | 0.84 | $0.82_{\pm0.00}$ | $\mathbf{0.78}_{\pm0.00}$ | 0.84 | $0.84_{\pm0.00}$ | $0.84_{\pm0.00}$ |
| | ECE | 9.75 | 5.35 | $5.15_{\pm0.13}$ | $4.16_{\pm0.13}$ | 2.66 | $1.87_{\pm0.05}$ | $\mathbf{1.77}_{\pm0.14}$ |
| | DECE | 9.37 | 5.34 | $4.99_{\pm0.12}$ | $4.08_{\pm0.09}$ | 2.33 | $1.69_{\pm0.18}$ | $\mathbf{1.39}_{\pm0.15}$ |
| | PEnt | 0.44 | 0.76 | $0.74_{\pm0.0}$ | $\mathbf{0.71}_{\pm0.01}$ | 1.01 | $1.00_{\pm0.00}$ | $1.02_{\pm0.02}$ |
| ImageNet | NLL | 0.97 | 0.96 | $0.94_{\pm0.00}$ | $\mathbf{0.93}_{\pm0.00}$ | 0.96 | $0.96_{\pm0.00}$ | $0.95_{\pm0.00}$ |
| | ECE | 4.66 | 3.31 | $2.25_{\pm0.06}$ | $1.62_{\pm0.06}$ | 2.54 | $1.71_{\pm0.04}$ | $\mathbf{1.60}_{\pm0.04}$ |
| | DECE | 4.65 | 3.36 | $2.25_{\pm0.03}$ | $\mathbf{1.45}_{\pm0.03}$ | 2.50 | $1.71_{\pm0.10}$ | $1.50_{\pm0.03}$ |
| | PEnt | 0.78 | 0.92 | $0.93_{\pm0.01}$ | $\mathbf{0.91}_{\pm0.01}$ | 1.06 | $1.05_{\pm0.01}$ | $1.01_{\pm0.00}$ |
| MNLI-m | NLL | 0.49 | 0.43 | $0.43_{\pm0.00}$ | $0.43_{\pm0.00}$ | 0.43 | $0.47_{\pm0.01}$ | $0.43_{\pm0.00}$ |
| | ECE | 7.52 | 1.43 | $1.09_{\pm0.13}$ | $\mathbf{0.88}_{\pm0.04}$ | 1.48 | $1.67_{\pm0.11}$ | $1.13_{\pm0.18}$ |
| | DECE | 7.53 | 1.37 | $0.63_{\pm0.05}$ | $\mathbf{0.35}_{\pm0.14}$ | 1.40 | $1.29_{\pm0.17}$ | $0.79_{\pm0.15}$ |
| | PEnt | 0.24 | 0.43 | $0.42_{\pm0.00}$ | $0.42_{\pm0.00}$ | 0.42 | $\mathbf{0.39}_{\pm0.00}$ | $0.40_{\pm0.01}$ |
| MNLI-mm | NLL | 0.47 | 0.42 | $0.42_{\pm0.00}$ | $0.42_{\pm0.00}$ | 0.42 | $0.45_{\pm0.01}$ | $0.42_{\pm0.00}$ |
| | ECE | 7.40 | 1.68 | $0.86_{\pm0.16}$ | $\mathbf{0.63}_{\pm0.05}$ | 1.62 | $1.64_{\pm0.13}$ | $1.20_{\pm0.10}$ |
| | DECE | 7.39 | 1.89 | $0.61_{\pm0.08}$ | $\mathbf{0.38}_{\pm0.21}$ | 1.91 | $1.52_{\pm0.23}$ | $0.10_{\pm0.6}$ |
| | PEnt | 0.24 | 0.42 | $0.41_{\pm0.00}$ | $0.41_{\pm0.00}$ | 0.42 | $\mathbf{0.39}_{\pm0.00}$ | $0.40_{\pm0.01}$ |

improve the calibration performance and do not yield significant overfitting. As a side note, we find that the choice of the loss function can substantially impact the behavior of the final calibrator: training with the NLL loss typically minimizes the test NLL well and improves the ECE by a reasonable amount, whereas training on the ECE loss typically minimizes the test ECE better (at least on the image tasks) at the cost of performing a little worse on the NLL and ECE.

We remark that the behaviors on MNLI are slightly different in that the best ECE is achieved by NRPT instead of NRPT+E, potentially due to the language task being different from the image tasks. However, we do observe that NRPT+E still performs best among those trained with ECE loss, again demonstrating the benefit of the improved expressivity in the NRPT calibrator.

**Comparison with deep ensembles; parameter efficiency** We further compare NRPT against deep ensembles (Lakshminarayanan et al., 2017) in Table 2. While deep ensembles in general can improve the accuracy and achieve much better NLL and sharpness (PEnt) due to the higher model capacity, we find that NRPT can consistently achieve a better ECE than an ensemble of 4 models, and nearly match an ensemble of 8 models on CIFAR100. NRPT further has much less memory overhead compared with the ensembles: the calibrators we used only has size $0.5\% - 2\%$ of the original model, whereas even an ensemble of 2 models requires doubling the model size. We also compare against Monte Carlo Dropout (Gal & Ghahramani, 2016) in Appendix C where we find Dropout cannot simultaneously maintain the accuracy and achieve calibration as well as NRPT.

## 4.2 CALIBRATION PERFORMANCE BEYOND THE ECE

**Accuracy among top-confidence predictions** We investigate the calibration ability of NRPT beyond the ECE metric, by looking at how well it improves the rank of the confidence among the individual examples. We measure the quality of the rank by visualizing the *accuracy among top-confidence predictions*: we take the subset of the test set for which the calibrated confidence ranks among the top $x\%$ (e.g. $20\%, 10\%$), and evaluate the accuracy (or error rate) among these examples. Roughly speaking, the error should become lower as we decrease the percentage $x$, but we can further compare this curve across different methods. In Figure 2a, we confirm that the NRPT has a lower classification error than TS and LTS for the majority of the percentages, except at the very tail. This can be further quantitatively measured by the PRR (prediction rejection ratio) metric, where a higher PRR implies a better accuracy among top-confident examples (see Appendix D for the details of PRR). In Table 3, we find that NRPT achieves better PRR than both TS and LTS.

Table 2: Comparison between NRPT and deep ensembles.

| Task | Metric | Uncal | NRPT | NRPT+E | Ens-2 | Ens-4 | Ens-8 |
|---|---|---|---|---|---|---|---|
| CIFAR100 | ACC | 80.36 | 80.36 | 80.36 | 82.42 | 83.56 | **84.10** |
| | NLL | 0.90 | 0.78 $_{\pm 0.00}$ | 0.84 $_{\pm 0.00}$ | 0.72 | 0.63 | **0.59** |
| | ECE | 9.75 | 4.16 $_{\pm 0.13}$ | **1.77** $_{\pm 0.14}$ | 4.77 | 2.63 | **1.71** |
| | DECE | 9.37 | 4.08 $_{\pm 0.09}$ | **1.39** $_{\pm 0.15}$ | 4.75 | 2.53 | **1.23** |
| | PEnt | 0.44 | 0.71 $_{\pm 0.01}$ | 1.02 $_{\pm 0.02}$ | 0.48 | 0.51 | 0.54 |
| | #params | 1x | 1.006x | 1.006x | 2x | 4x | 8x |
| ImageNet | ACC | 76.27 | 76.27 | 76.27 | 77.29 | **77.76** | - |
| | NLL | 0.97 | 0.93 $_{\pm 0.00}$ | 0.95 $_{\pm 0.00}$ | 0.90 | **0.87** | - |
| | ECE | 4.66 | 1.62 $_{\pm 0.06}$ | **1.60** $_{\pm 0.04}$ | 2.43 | 1.79 | - |
| | DECE | 4.65 | **1.45** $_{\pm 0.03}$ | 1.50 $_{\pm 0.03}$ | 2.31 | 1.73 | - |
| | PEnt | 0.78 | 0.91 $_{\pm 0.01}$ | 1.01 $_{\pm 0.00}$ | 0.83 | 0.88 | - |
| | #params | 1x | 1.015x | 1.015x | 2x | 4x | - |

Table 3: Prediction rejection ratio (PRR) metric (def in Appendix D). Higher the better.

| Task | Uncal | TS | LTS | **NRPT** |
|---|---|---|---|---|
| CIFAR100 | 56.49% | 54.28% $_{\pm 0.00\%}$ | 60.69% $_{\pm 0.22\%}$ | **61.84%** $_{\pm 0.37\%}$ |
| ImageNet | 56.11% | 55.69% $_{\pm 0.00\%}$ | 58.47% $_{\pm 0.14\%}$ | **58.65%** $_{\pm 0.07\%}$ |

**Tradeoff between sharpness and ECE** For post-calibration methods that maintain the accuracy of the model, the calibration (e.g. ECE) is typically negatively correlated to the sharpness of the prediction (e.g. predictive entropy). We investigate the ability of NRPT in optimizing the tradeoff when both metrics are desired. To test this, we train both {TS, LTS, NRPT} on a weighted combination of the NLL and ECE loss $\alpha L_{\text{NLL}} + C(1 - \alpha) L_{\text{ECE}}$, where use multiple weight values $\alpha \in \{0, 0.1, \ldots, 0.7, 0.75, 0.8 \ldots, 1\}$, and plot the resulting predictive entropy and ECE as a tradeoff curve. In Figure 2b, we see that NRPT achieves a nearly universally better tradeoff curve than TS and LTS. This suggests that the improved expressivity in NRPT can be beneficial in practice when more than one metrics are desired and it is necessary to manage the tradeoff.

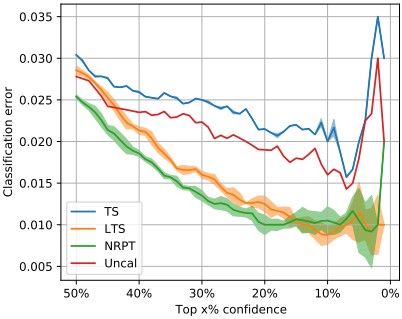

(a) Error within top-confidence predictions

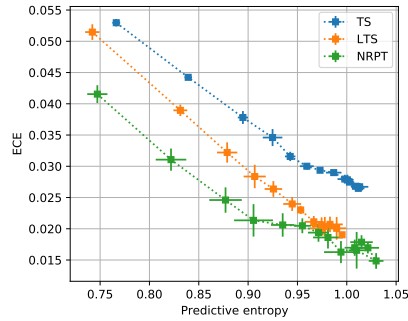

(b) Tradeoff between ECE and sharpness

Figure 2: Comparison between TS, LTS, and NRPT in the calibration abilities. Each dot in (b) is obtained by optimizing a weighted combination of the NLL and ECE loss. Shaded area in (a) and crosses in (b) indicate the standard deviation over 4 random seeds.

## 5 CONCLUSION

We proposed Neural Rank-Preserving Transforms (NRPT), an improved technique for uncertainty post-calibration, and showed that it outperforms existing post-calibration methods on benchmark tasks. A number of interesting research questions remain open: for example, can we have a better understanding on the choice of the architecture for the monotonic transforms used in NRPT? Can we build calibrators that combine the advantages in standard post-calibration methods and ensemble-like methods? We would like to leave these as future work.

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

## A    ADDITIONAL RELATED WORK

**Post-calibration methods**    Post-calibration for binary classification problems has been studied extensively in classical work, including non-parametric (bining type) methods such as histogram binning (Zadrozny & Elkan, 2001), isotonic regression (Zadrozny & Elkan, 2002), Bayesian binning into quantiles (Naeini et al., 2015), as well as parametric methods such as Platt scaling (Platt et al., 1999) which re-fits a one-dimensional logistic regression model from the logit to the probability on a holdout calibration dataset. Kumar et al. (2019) combines the advantage of non-parametric and

parametric approachs in the scaling-bining calibrator, which first fits a parametric function to the calibration dataset and then performs the binning.

While most binning type methods are defined for binary problems, they can be extended to multi-class classification ($K \geq 3$ classes) by performing the calibration on all the 1-vs-$K-1$ binary tasks, and re-normalizing the calibrated probabilities (Zadrozny & Elkan, 2002). The parametric can also be extended to the multi-class case with various degrees of freedom, including tempreature scaling, vector scaling, and matrix scaling. Guo et al. (2017) tested the multi-class calibration methods on a variefy of tasks and found that temperature scaling performs the best across the board. Local Tempreature Scaling (Ding et al., 2020) proposes to use an individualized temperature for each example; in this paper we implement . Dirichlet calibration (Kull et al., 2019) improves the per-class calibration by using a different Dirichlet distribution for each class as the calibrator. MMCE (Kumar et al., 2018) proposes to optimize a kernalized version of ECE for improving the calibration; however they do not consider optimizing the original ECE directly.

**Diversity-based uncertainty quantification**  Diversity-based uncertainty quantification can be roughly divided into two types. *Ensemble methods* such as deep ensembles (Lakshminarayanan et al., 2017) train an ensemble of models from different initializations (and with different SGD noise), and find that the aggregated (average) predicted probability exhibit better uncertainty calibration than a single deterministic model. As ensembles are memroy and runtime heavy, a recent line of work proposes to make ensembles more efficient by either reducing the parameter count through smart reparametrizations (Wen et al., 2020) or a single deterministic model that simulates the ensembles (Liu et al., 2020). A related line of work proposes to distill an ensemble of models (Malinin & Gales, 2018; Malinin et al., 2020; Tran et al., 2020). We remark that either the efficient ensembling approach or the distillation approach improves the uncertainty calibration through simulating an ensemble, and can be used jointly with post-calibration methods.

*Bayesian neural networks* (MacKay, 1995) are capable of producing uncertainty estimates by nature since it learns a distribution of networks (that can be used aggregatedly) rather than a single network. Monte Carlo Dropout (Gal & Ghahramani, 2016) uses the randomized prediction capability of Dropout to perform uncertainty calibration. SWAG (Maddox et al., 2019) performs uncertainty calibration via an approximate Bayesian model averaging using the SGD iterates. Bayesian rank-one factors (Dusenberry et al., 2020) is a Bayesian version of BatchEnsembles that learns a posterior over the rank-one parametrization of ensembles.

## B  ADDITIONAL EXPERIMENTAL DETAILS

### B.1  MODELS

**NRPT**  We choose $\sigma$ in (7) to be the ReLU activation and $\phi$ to be the leaky relu activation with negative slope tuned in $\{0.5, 0.8, 1.5, 2.0\}$. The number of neurons $M$ was tuned in $\{5, 10\}$. We further initialize the $T_{\theta_j}(x)$ such that it has initial values (approximately) $\{0.5, 1.0, 1.5, \ldots, 0.5M\}$ by properly initializing the bias within these networks.

**LTS**  For LTS (local temperature scaling) we use an architecture that is similar to NRPT:

$$T_\theta(x) = \sigma_{\text{temp}}(a^\top \sigma(W\widehat{H}(x) + b)),$$

where $H : \mathbb{R}^d \to \mathbb{R}^{d_{\text{hid}}}$ is the last hidden representation layer of the trained classifer, $M \in \mathbb{R}^{d_{\text{hid}} \times N}$ and $a \in \mathbb{R}^N$. Similar as in NRPT, we chose $N = 512$ and $\sigma_{\text{temp}}(t) = 0.2 + \text{relu}_6(t)$. We remark that our implementation is likely different from the implementation in (Ding et al., 2020) (and operates on different base models). Nevertheless we find that our implementation is also a strong calibrator that consistently performs temperature scaling.

### B.2  TRAINING AND EVALUATION

**CIFAR-100**  The base WideResNet-28-10 on CIFAR-100 was trained with batchsize 128 for 200 epochs with a cosine learning rate with initial learning rate 0.1.

**ImageNet** The base WideResNet-50-2 on ImageNet was trained with batchsize 256 (parallelized onto 8 GPUs) for 100 epochs. The initial learning rate was 0.1, with a fixed decay of factor 0.1x at the $\{30, 60, 80\}$-th epochs.

**MNLI** The BERT-base on MNLI was finetuned from the pretrained model with batchsize 32 for 3 epochs, with the AdamW optimizer and learning rate $2 \times 10^{-5}$.

All post-calibrators are trained with a one-cycle learning rate (Smith, 2017) and we tune the initial learning rate within {1e-3, 3.1e-3, 1e-2, 3.1e-2}. All post-calibrators are trained with the same batchsize as used in training the base model. The number of epochs for training the calibrators was 50 on CIFAR-100, 5 on ImageNet, and 6 on MNLI.

**ECE as loss and evaluation metric** We choose the number of bins to be 15 for evaluating the ECE following the standard practice in (Guo et al., 2017) (and the body of recent work). However, at train time, we tune the number of bins within $\{5, 10\}$, as we evaluate the train-time ECE loss on small minibatches, which could benefit from a smaller number of bins.

**Hyperparameter tuning** All the hyperparameter tuning were conducted by further splitting the calibration dataset into a training and development set, where we train with a grid of hyperparameters on the train set and select the best on the development set.

## C  COMPARISON BETWEEN NRPT AND MONTE CARLO DROPOUT

We compare the calibration performance of NRPT and Monte Carlo Dropout (Gal & Ghahramani, 2016) on CIFAR-100, where we train a WideResNet-28-10 model for each drop probability, and evaluate the calibration by aggregating over 8 random predictions at test time (each with a different mask).

We observe a consistent trend on Dropout: increasing the drop probability improves the ECE at the cost of hurting the accuracy, which is as expected since randomized predictions can naturally get more calibrated but less accurate as we increase the level of randomization. Comparing NRPT+E with Dropout, we see that NRPT+E achieves better ECE than Dropout up to drop probability 0.7; for drop probability 0.8, the ECE is better than NRPT+E, however it comes at the cost of significantly lower accuracy. For the NLL and predictive entropy metric, NRPT performs slightly better than Drop 0.8 and slightly worse than Drop 0.7. However, Drop 0.7 also has a worse accuracy than NRPT (which did not change the accuracy of the model). This suggests that NRPT may be preferred over Dropout if maintaining the accuracy is crucial; Dropout can only achieve a better calibration by huring the accuracy.

Table 4: Comparison between NRPT and Monte Carlo Dropout. "Drop $0.4$" indicates the dropout method with drop probability $0.4$ and keep probability $0.6$. All dropout methods are evaluated by aggregating the randomized predictions over 8 masks.

| Task | Metric | Uncal | **NRPT** | **NRPT+E** | Drop 0.4 | Drop 0.6 | Drop 0.7 | Drop 0.8 |
|---|---|---|---|---|---|---|---|---|
| | ACC | 80.36 | 80.36 | 80.36 | 80.91 | 80.21 | 78.92 | 77.61 |
| | NLL | 0.90 | **0.78** $_{\pm 0.00}$ | 0.84 $_{\pm 0.00}$ | 0.75 | **0.74** | 0.76 | 0.81 |
| CIFAR100 | ECE | 9.75 | 4.16 $_{\pm 0.13}$ | **1.77** $_{\pm 0.14}$ | 4.91 | 3.03 | 2.55 | **1.01** |
| | DECE | 9.37 | 4.08 $_{\pm 0.09}$ | **1.39** $_{\pm 0.15}$ | 4.71 | 2.96 | 2.19 | **0.50** |
| | PEnt | 0.44 | 0.71 $_{\pm 0.01}$ | 1.02 $_{\pm 0.02}$ | 0.58 | 0.63 | 0.67 | 0.76 |
| | #params | 1x | 1.006x | 1.006x | 1x | 1x | 1x | 1x |

## D  DETAILS ON THE PRR METRIC

The PRR (prediction rejection ratio) metric is commonly used for quantitatively summarizing the accuracy against top-confidence predictions, or equivalently for evaluating the if the model can reliably reject to predict (Malinin et al., 2020). The PRR metric is based on the AUC (area under curve) on the accuracy among top-confidence curve as in Figure 3. We compare the curve of a method

against that of an "oracle" rank of confidence (which perfectly ranks all wrong predictions of lower confidence than all correct predictions), as well as a "random" rank of confidence illustrated by a straight line. The PRR metric is then defined as

$$\mathrm{PRR(Method)} := \frac{\mathrm{AUC(Random)} - \mathrm{AUC(Method)}}{\mathrm{AUC(Random)} - \mathrm{AUC(Oracle)}} = \frac{\text{shaded area in green}}{\text{shaded area in orange}}.$$

A higher PRR indicates that the method achieves a better (closer to the oracle) rank of confidence.

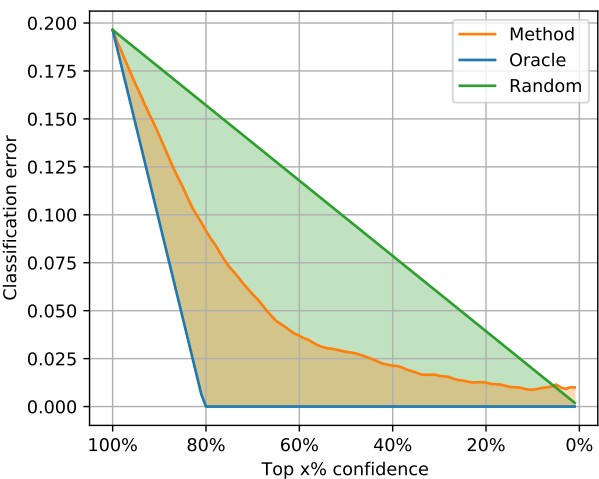

Figure 3: Illustration of the PRR metric.

# E ACCCURACY AMONG MOST-CONFIDENT EXAMPLES ON IMAGENET

We plot the accuracy among most-confident examples on ImageNet in Figure 4. Similar as on CIFAR-100 (Figure 2a), we see that NRPT achieves lower error rate than LTS and TS at most confidence levels, except for the very tail.

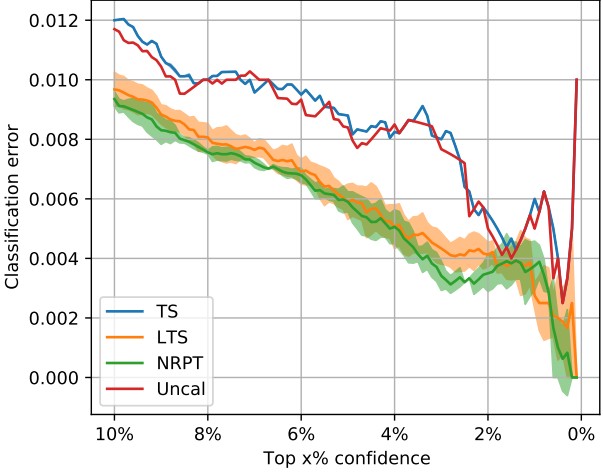

Figure 4: Accuracy among most-confident examples on ImageNet.

