# OpenReview forum: "Improved Uncertainty Post-Calibration via Rank Preserving Transforms"
_ICLR.cc/2021/Conference — Reject_

### Official Review · AnonReviewer1 · 2020-10-28
**Interesting work, some merits as well as weaknesses**

**Rating:** 5
**Confidence:** 4

**Review:**

**Summary**
This work proposes a method for calibrating outputs of deep neural networks using higher capacity learning than previously possible. This is accomplished by a “Neural Rank Preserving Transform” (NRPT) which is applied post-logit to the network outputs. The learned NRPT transform is monotonic, it preserves the rank of the probabilities between logits. The authors also propose to train using a loss for the Expected Calibration Error (ECE), which is the metric that calibration models seek to minimize. This is a straightforward way to improve model calibration, and to my knowledge it has not been proposed in prior works.

**Novelty**
The are reasonably solid novel contributions in this work, to my estimation. The insight as to why high-capacity calibration methods overfit and show poor ECE, which leads to the insight to preserve the accuracy of the model via enforcing monotonicity. As mentioned above, the use of ECE as a loss term also appears to be novel.

**Impact**
Calibration of neural networks is an important issue which is of concern to the ML community at large, especially in sensitive applications such as medical diagnostics and autonomous driving.


**Clarity**
The paper is reasonably clear and well written. However there are some strange organizational choices, such as the lack of a clear related work section. This has led to the omission of several baseline methods that should have been cited and compared against. Some details of the experimental setup are not clear. For example, exactly how the train/calibration/val splits are used are not clear. One can assume, but it should not be necessary.

**Evaluation**
The evaluation seems reasonable, appropriate metrics and datasets are chosen, however I am not sure how much insight about calibration we can really glean from NLL and PEnt. These, to me, are more indirect measures of calibration and more importance should be given to the ECE and DECE results. As the authors point out, it is curious that the +E models are outperformed by the vanilla models on the MNLI datasets. Since this happened on ½ of the datasets, I would have liked to see more datasets to verify that this is truly an anomaly. I’m not sure why other classic calibration techniques have not been considered such as isotonic regression, platt scaling, and Bayesian Binning.


**Strengths (Reasons to accept)**
The novel contributions mentioned above are of interest to the community.

Experiments are reasonably convincing although lacking in some respects.


**Weaknesses (Reasons to reject)**
My big problem with this paper is the use of the word “uncertainty”. In my view, this paper is very clearly about calibration which is concerned with the problem of making sure the model outputs match true accuracies. Uncertainty estimation is concerned with determining the epistemic and aleatory uncertainty. Many uncertainty estimation methods can, as a side-effect, improve calibration. But this is typically not their aim. I do see some minor value in showing the results in Figure 3, but I take issue with saying the method performs “uncertainty post-calibration”.

Another criticism is that the authors do not acknowledge that other calibration methods such as temperature scaling have the desirable property of being monotonic not just in terms of the logits, but in terms of the rank of the confidences. That is, the order of the argmax classes will be preserved. To my understanding, the proposed method does not explicitly ensure this.

Finally, as mentioned above, there are some weaknesses in the evaluation.


=============== UPDATE ==============

After reading the concerns of the other reviewers and the author response, it seems that many of the concerns remain unaddressed especially the top concerns. Accordingly, I have decided to lower my score.

---

### Official Review · AnonReviewer3 · 2020-10-28
**New post-processing high-capacity calibration method that is constrained to main classification ranking**

**Rating:** 7
**Confidence:** 4

**Review:**

This paper introduces a new post-processing calibration method that can keep classification ranking, like the simple (local) temperature scaling methods, but has larger capacity, comparable to matrix scaling.  Essentially it tries to achieve the best of both approaches by avoiding the underfitting and overfitting issues of temperature and matrix scaling, respectively.  The method proposed essentially learns a temperature and a bias parameter that depends on the image, and two matrices that are global to all images.  During post-processing calibration training, the constraining of the matrices and the temperature to be strictly positive guarantees that the calibration is monotonically increasing, guaranteeing the preservation of classification ranking.  The paper further shows that this calibration model can be trained in a post-processing approach with NLL and ECE losses (the training with the ECE loss is claimed as another novelty).  Experiments are performed on CIFAR100, ImageNet, and MNLI, and results show that the proposed post-processing calibration stage trained with NLL improves NLL and Entropy for CIFAR100 and ImageNet, and the one trained with ECE improves the ECE related measures.  For MNLI, results seem to show that the calibration with NLL loss provides the best general result.  Additional results that show the value of the method for other calibration measures also display the value of the proposed approach.

I find the paper very easy to read, with clear claims and clear explanations.  The method proposed is quite simple and extends local temperature scaling marginally from a technical viewpoint.  Nevertheless, I like the simplicity of the approach and the experiments that show the potential value of the approach.  In general, I think it is a good paper and would support its acceptance.

Minor issues:
1- The strange results of NRPT+E on MNLI-m and MNLI-mm for the ECE measures need to be better explained since they are quite unintuitive.
2- The result of NRPT+E on MNLI-mm for DECE is 0.10, which is best for the row -- that should be the boldface number for that row.
3- In section 4.2, the paper says "In Figure 2a, we confirm that the NRPT has a lower classification error than TS and LTS for the majority of the percentages, except at the very tail."  This needs to be explained.

---

### Official Review · AnonReviewer2 · 2020-10-28

**Rating:** 2
**Confidence:** 5

**Review:**

**Justification for Score**
The paper is missing a thorough literature review and there are many missing citations and comparisons. Overall, unfortunately there are multiple misleading (and sometimes false) claims throughout the paper. The evaluation performance is also a little unfair as they use ECE as an objective and then only show better performance on ECE. Overall, the paper is well written but the content does not have good enough quality as there seem to be many unsupported claims.

## Review

### Summary
This paper proposes a post-hoc calibration method which aims to preserve the accuracy of the classifier as well as improve its uncertainty calibration. The core of the method lies in formulating a general form of the commonly used scaling method Temperature Scaling (TS) and more recent variation of it, Local TS (Local TS). The proposed method Neural Rank-Preserving Transforms (NRPT) maintains the accuracy but also shows better calibration performance compared to TS and Local TS.

### Strengths
* It is important that a post-hoc calibration does not decrease the accuracy. I like the justification of the authors about this: " a calibrator that does not maintain the accuracy may attempt to improve the accuracy at the cost of hurting (or not improving) the calibration". Though, I have to point out that a method which can change the rank performance also has advantages compared rank-preserving methods (i.e. TS or NRPT): The literature has shown that accuracy and calibration performance can jointly be improved - the authors don't really acknowledge this point (see below for references and examples of such methods).

* A rank preserving method has the advantage of optimizing loss functions which do not have to "care" about accuracy. In this case, the authors propose to use the ECE loss. A method which does not preserve the rank would completely decay the accuracy in an attempt to optimize the ECE. That being said, I still think that optimizing on the ECE and only showing improvements on the ECE metric is not surprising (see below for more discussion on this point). So it does have the advantage of optimizing ECE, but the paper does not really show the benefit here as it only shows that optimizing the ECE performs better on ECE (this is expected) but performs worse on the other metrics (see Tab. 1). Also, I would like to point out that I include DECE when I talk about ECE in this review.

* Better calibration performance compared to the Baseline, TS and LTS. The authors also compared against deep ensembles which have been shown to improve calibration, though they only perform better on ECE for the case where NRPT is optimized using the ECE (NRPT-E). NRPT (without -E) performs significantly worse than ensembles larger than 4 (at least more than half better).

* Even though NRPT does not have overall better performance compared to deep ensembles (i.e. only is better at ECE metric and only when ECE is specifically optimized during training), it does have the advantage of being computationally faster.


### Weaknesses
1. The major weakness of this paper is its view of the current post-hoc calibration literature seems to be outdated. A re-occurring theme throughout the paper is that post-hoc methods are very simple and always talks about Temperature Scaling (TS) as an example. The post-hoc literature has come far beyond simple methods such as TS. Even though it is true that many recent variants of TS have shown good improvements, there are many other non-"simple" methods which have much (1) larger capacity, (2) can improve accuracy (i.e. they do not have accuracy loss) and (3) greatly improve calibration performance.

2. The related work paragraph also only talks about three post-hoc calibration methods. Even though, the literature grows very fast and it can be hard to always keep track of all new papers, there have definitely been much more than the three variants cited in this paper (see below for references). Later on, the paper does cite the work [1] and claims that they show that "overfitting cannot be easily fixed by applying common regularizations such as L2 on the calibrator". Despite, this statement not being entirely true (see below), it seems that the authors are aware of this paper. [1] has also presented a calibration method (i.e. Dirichlet calibration - mentioned in the title of [1]), so why has this method not been compared or cited for its calibration method?
[1] has thoroughly compared against multiple other calibrators (in addition to TS and Matrix scaling), so why were none of these other calibrators cited, used and compared against?

3. This is why I find that there are many misleading statements throughout the paper which imply that the (large) family of post-hoc calibration methods can be summarized as "simple" and limited to TS or Matrix Scaling. One example of such a statement can be found in the appendix: "Existing post-calibration methods such as temperature scaling recalibrate a trained model using rather simple calibrators with one or few parameters, which can have a rather limited capacity." The reason I find them misleading is that for a reader/reviewer who is not familiar with post-hoc calibration, it might seem that TS (and its variants) are the only and best methods for post-hoc calibration. Recent years have shown multiple other solutions which are not limited to such simple methods.

4. Here is a list of some recent and relevant approaches: GP [3], Histogram Binning [5], I-Max histogram binning [4], Beta Calibration [2], Dirichlet Calibration [1], Matrix Scaling with ODIR regularization [1], BTS [6], Isotonic Regression [9] and MnM [8]. These post-hoc calibration methods range from relatively old methods to much newer methods. Even though I certainly don't expect to see all of them cited and compared against, the authors should do a much more thorough search of the literature or not make statements which imply that the post-hoc literature is only limited to the methods which they compare against.

5. Even though the authors compared against deep ensembles, the performance is only better in one case: the ECE metric and only when NRPT optimizes the ECE loss (i.e. NRPT-E). This is no surprise and cannot be used to claim that the method is better than deep ensembles. A method which optimizes the metric A directly will perform better at this metric A. Of course, there are some exceptions such as NLL training without regularization. But overall this is not really an impressive result, especially given that the ECE has lately been criticized (see above).

6. State-of-the-art-performance: "Local Temperature Scaling (LTS, see (4)), a generalization of temperature scaling that is observed to achieve state-of-the-art performance on a variety of computer vision tasks (Ding et al., 2020)" This paper claims that LTS obtains state-of-the-art performance and by performing better than LTS implying that they have beaten state-of-the-art in post-hoc calibration. The authors of this current paper do not claim that they obtain state-of-the-art but they imply this by mentioning that they perform better than a method which is previously state-of-the-art. I looked into the LTS paper and they also did not make the claim of obtaining state-of-the-art performance. So where is this claim coming from? It is hard to claim that LTS obtains state-of-the-art performance. After a brief look in the LTS paper, it seems that they too ONLY compare against TS variants (please correct me if I am wrong). As I have listed above, there are multiple other methods which have shown to improve calibration and have a thorough comparison against multiple other post-hoc calibration methods (and not only limited to TS and some of its variants).  Again, I find this a little misleading that the authors of this paper seem to imply many statements to make their method seem good.

7. Fig. 1 and Matrix Scaling: The authors seem to want to show the benefit of NRPT by showing superior performance compared to TS ("simple") and Matrix Scaling ("complex calibrators can often overfit"). Despite, the fact that these are poor baselines to compare against (see list of calibrators above), there has been a new proposal to fix the overfitting issues of Matrix Scaling. [1] proposed to use ODIR regularization ("simple L2 regularization") to improve the performance of Matrix Scaling. Firstly, the authors cite this work which means they are aware of this paper and this technique to reduce the overfitting. So then why has it not been used to in Fig. 1? And why was it chosen to compare against Matrix Scaling (without regularization) if there does exists a better solution for matrix scaling? Secondly, the authors instead make a misleading statement: "It is further observed that the overfitting cannot be easily fixed by applying common regularizations such as L 2 on the calibrator (Kull et al., 2019)" This statement is not true. This paper shows in Table 3 (of [1]) and Table 4 (of [1]) that Matrix Scaling with ODIR (i.e. L2 regularization) is sometimes the best performing method. Even for the rest of the cases where it does not perform best, it performs similarly. This shows that "simple regularization" can help address the over-fitting issue. This regularization has also been used in [4] with Matrix Scaling for the ImageNet classifiers and has shown to be the best performing method in terms of accuracy (showing that it helps with reducing the overfitting) Again, this is a mis-leading statement and the literature does not support this.

8. Based on the previous (possibly) false claim, the next claim that "This empirical evidence seems to suggest that complex calibrator with a a large number of parameters are perhaps not recommended in designing post-calibration methods" is also not entirely true. As mentioned above, highly expressive methods can improve calibration.

9. The authors mention that the small size of the validation set can cause highly expressive methods to overfit, thought [3,4] have both shown to greatly improve calibration performance with as little as 1000 samples for ImageNet classifier calibration and even shows better performance than methods using significantly more samples. Therefore, it is not true that a limited dataset will lead to overfitting and these results show that regularization and other techniques can handle a low data regime for calibration.

10. The authors also claim that "matrix scaling is not guaranteed to maintain the accuracy". This claim is true, but the authors fail to mention that Matrix Scaling (and its regularized variants) can also improve the classification accuracy performance. The high capacity allows it to improve accuracy as well. After using L2 regularization the accuracy performance can be greatly improved.
Alternatively, [5], also presented Vector Scaling (which has less capacity than Matrix Scaling but more than TS) and they also show that it too can change the accuracy performance and actually improve the accuracy in most cases. So even though I do see the benefit of having method which preserves the rank, it should clearly be pointed out that it has the disadvantage of not being able to improve the accuracy. For example, compared to [3] which improves the accuracy performance, NRPT has the disadvantage with respect to accuracy. So despite this above statement being true, the authors use this statement to justify why methods like TS which preserve the accuracy are better, though TS actually has a disadvantage compared to the many cases where these rank-changing methods improve the accuracy.

11. ECE: The authors cite [10] to use their presented debiased estimator. Though, they completely ignored the main message of that paper. [10] has shown that continuous output scaling methods (e.g. TS) have trouble estimating the ECE and instead propose to use binning or quantized output methods instead. [1] also shows how the ECE is underestimated when using too few bins during evaluation. They show that the ECE of scaling methods are non-verifiable. So even though the authors are aware of this paper, they do not at all comment on the use of ECE for continuous output scaling methods such as the one presented. Other recent works [3,4] have also discussed this and proposed solutions such as increasing the number of bins when estimating the ECE of scaling methods. How many bins were used to estimate the ECE? As the authors are aware of this work, they should discuss this weakness of ECE in their paper and more importantly use approaches used in other recent works in the literature to address these issues to some extend. Again, it seems that the authors need to do a more thorough literature search on post-hoc calibration.

12. The method also uses ECE during training: Even though this might be an advantage of rank preserving methods, it is not surprising that the method using ECE as a loss performs the best at ECE. Table 1 shows that "NRPT+E" is only better at ECE and no other metric. As discussed earlier, the ECE metric has major flaws and problems, so using this as a loss also comes with its problems. So I do not see this variant of NRPT as very useful. It would be interesting to how "NRPT-E" performs on other metrics and tasks which are not directly optimized during training, as using ECE as the objective and the evaluation is unfair.

13. Tab.1 : Why are the uncalibrated ECEs so high? These are unusually high ECE even for uncalibrated networks? There is no surprise then that even simple methods such as TS can show great improvement when the networks are so miscalibrated. Many other works using similar networks report much lower uncalibrated ECE. Maybe the ECE was evaluated differently?

14. ECE as objective: As you use ECE during the learning, you should be providing all details about ECE (e.g. the number of bins, the bin widths or bin edges ).

### Minor comments
* Some references as using arxiv versions of the paper (e.g. Guo2017 is a ICML paper from 2017, so shouldn't be using an arxiv reference)


### References (all can be found on arxiv)

[1] Meelis Kull, Miquel Perello Nieto, Markus Kängsepp, Telmo Silva Filho, Hao Song, and Peter Flach.  "Beyond temperature scaling: Obtaining well-calibrated multi-class probabilities with dirichlet calibration"

[2] Meelis Kull, Telmo Silva Filho, and Peter Flach. "Beta calibration: a well-founded and easily implemented improvement on logistic calibration for binary classifiers".

[3] Jonathan Wenger, Hedvig Kjellström, and Rudolph Triebel. "Non-parametric calibration for classification"

[4] Kanil Patel, William Beluch, Bin Yang, Michael Pfeiffer, Dan Zhang. "Multi-Class Uncertainty Calibration via Mutual Information Maximization-based Binning"

[5] Chuan Guo, Geoff Pleiss, Yu Sun, and Kilian Q Weinberger. "On calibration of modern neural networks"

[6] B. Ji, H. Jung, J. Yoon, K. Kim, and y. Shin. Bin-wise temperature scaling (BTS): "Improvement in confidence calibration performance through simple scaling techniques"

[7] Jize Zhang, Bhavya Kailkhura, and T Han. "Mix-n-Match: Ensemble and compositional methods for uncertainty calibration in deep learning."

[8] Mahdi Pakdaman Naeini, Gregory F. Cooper, and Milos Hauskrecht. "Obtaining well-calibrated probabilities using Bayesian binning."

[9]  Bianca Zadrozny and Charles Elkan. Transforming classifier scores into accurate multiclass probability estimates.

[10] Ananya Kumar, Percy S Liang, and Tengyu Ma. "Verified uncertainty calibration"

---

> ### Comment · AnonReviewer4 · 2020-11-15
> **These points need to be addressed.**
>
> Thank you for writing such a thorough review! Not being familiar with the calibration literature, I came away with a very different impression. However, I found your points quite convincing - I plan on decreasing my score unless the authors offer a compelling response.

---

> > ### Comment · AnonReviewer1 · 2020-11-15
> > **I agree**
> >
> > I concur with AnonReviewer4. I would like to see the authors respond to these comments.

---

> ### Author Response · Authors · 2020-11-25
> **Response**
>
> Thank you for the very thoughtful and detailed review of our work. We agree with many of the limitations you pointed out, such as comparison with certain post-calibration techniques (1-4), matrix scaling + ODIR regularization as another way of reducing the overfitting (7-10). We appreciate all your comments and will incorporate them into the next version of this work.
>
> We would still like to make a few clarification points about the review:
>
> -- Optimizing the ECE and only shown performance on ECE (comments 5, 12)
>
> We did not claim our strength solely based on our results on optimizing the ECE. In fact, we also optimized the NLL, and achieved better NLL & ECE than the existing methods using the NLL as a loss function (Table 1). Also, we showed that we can optimize a linear combination of the NLL and ECE and obtain a tradeoff curve that is everywhere better than temperature scaling or local temperature scaling.
>
> Further, our NRPT is able but not limited to using the ECE as a loss function, and we see that as our strength rather than weakness: it offers an option, rather than restricts the user to only use one particular loss function. If a practitioner cares about both sharpness/NLL and ECE, then he/she can use a combination of the loss function.
>
> -- Our uncalibrated ECEs seems too high (14)
>
> Our uncalibrated ECE is about the same as reported in the literature. For example, our uncalibrated ECE on CIFAR100 is 9.75%. Guo et al. [5] reported 10% - 16% for four different architectures on the same dataset (their Table 1). On ImageNet our uncalibrated ECE is 4.66%, [5] reported 5% - 6% on the same dataset; Ovadia et al. reported 3%-4% (their Figure 2).
>
> -- Missing details of ECE in training (13)
>
> We provided the details of the ECE in Appendix B.2, “ECE as loss and evaluation metric”. The number of bins is {5, 10} for training and 15 for evaluation. The bins are equally spaced as described in Section 2.

---

### Official Review · AnonReviewer4 · 2020-10-28
**Interesting and thorough work!**

**Rating:** 4
**Confidence:** 3

**Review:**

Overview:

This paper proposes a post-calibration technique that is meant to be more powerful than temperature scaling without introducing overfitting. The authors argue that previous attempts to generalize temperature scaling have a tendency to overfit not because of the additional parameters, but rather because they are not rank preserving (and therefore can change model predictions / accuracy). The key idea is to use a higher-capacity (2-layer neural network) calibrator that is constrained to be rank-preserving so that model accuracy remains unchanged. The proposed method is compared against baselines on classification tasks from vision (CIFAR-100, ImageNet) and language (MNLI).

Strengths:

The introduction is interesting reading, and does a great job motivating the importance of model calibration and putting the paper in context. Generally speaking, the paper is quite readable and the figures/tables are clear and informative. The proposed approach is simple, reasonable, and somewhat surprising - it would be easy to assume that higher capacity calibrators fail due to high parameter count, but the paper argues that rank-preservation is really the key. This is interesting and (as far as I know) novel.

The experiments are very thorough - all metrics are averaged over multiple runs and several complementary performance metrics are provided. The proposed method beats the baselines in most (but not all) of these cases. I think the authors make a convincing case for their approach, but they also successfully motivate further exploration into higher-capacity calibrators by showing that overfitting is not inevitable.

Weaknesses:

The paper relies almost exclusively on summary metrics and does not conduct too much fine-grained introspection. For instance, the authors observe peculiar behavior for their proposed method on high-confidence examples but do not explain why. There could also be more effort invested in discovering why the proposed method outperforms the competition. For instance - how is calibration improvement distributed over examples? Do the performance gains come from reducing relatively few gross errors or from improving many moderate errors? Are there examples that are actually calibrated worse with the proposed method? While space constraints limit what can be included in the main paper, I think this kind of deeper analysis would improve the work.

Overall:

While I'd like to see some deeper analysis, I think this is paper makes an interesting and useful contribution to the calibration literature, backed up by solid empirical results.

Minor comments:

Section 1: "a much less number of additional parameters" should be replaced by something like "a much smaller number of additional parameters".

Section 3: In the "Instantiation via monotone two-layer networks" section, it would be helpful for clarity if $a_j$, $\theta_j$, and $b_{\theta_j}$ are explicitly described. Similarly, in the "Architectural choices" section it is not entirely clear which parameters $\theta_1, \ldots \theta_M$ are each supposed to represent. Also, what is $\sigma$ (as opposed to $\sigma_\mathrm{temp}$ and $\sigma_\mathrm{SM}$)? Though this is described later, the reader may not want to jump around to hunt down these symbols.

Section 4.1: It would have been nice to see the vision results on a more standard architecture (e.g. vanilla ResNet-50) which could have been used for both vision tasks.

Figure 2: The caption should specify what dataset we're looking at.

Appendix B.1: I think "consistently performs temperature scaling" should be "consistently outperforms temperature scaling" under the LTS heading.

**UPDATE AFTER DISCUSSION:** R4's exceptionally thorough review raised a number of important concerns which I initially did not recognize. Moreover, the most important of these concerns went almost entirely unaddressed by the authors. Since the paper has not been appropriately revised, I have decided to lower my score from a 7 to a 4.

---

### Public Comment · ~Jize_Zhang1 · 2020-11-13
**One related paper & the training objective of NRPT**

Thanks for the great work on designing NRPT calibration method! I also strongly agree on the importance to consider accuracy-preserve feature. In fact, we have a very related paper targeting at the same desiderata (calibrating while keeping the model accuracy unchanged) that I highly encourage the authors to discuss and compare with:

Jize Zhang, Bhavya Kailkhura, and T Han. "Mix-n-Match: Ensemble and compositional methods for uncertainty calibration in deep learning.", ICML 2020

Specifically, you will see that the "Rank Preserving Transforms" in Section 3 is very similar to  the "Accuracy-preserving Calibration" in the Mix-n-Match paper Section 3.1. The only difference is that it enforces monotonicity on the softmax probabilities before/after calibration and in this work, it is done on logits. But both conditions are equivalent.

With respect to the training of neural networks calibrators on ECE: the danger of using ECE as the training objective would be the introduction of histogram estimation noise and the non-smoothness. For the proposed NRPT approach, is it better to just use mean squared error as the objective function? Because in Sec 4.2 of Mix-n-Match, we showed that minimizing MSE for accuracy-preserving methods (including NRPT in this case) is equivalent to minimizing the squared ECE, but now the objective function is binning-free and easy-to-optimize. Or if your ultimate goal is to minimize top-label (confidence) ECE, just use MSE w.r.t top labels as the objective.

I also highly recommend to check out our kernel-based ECE estimator in Mix-n-Match as an alternative choice to the histogram-based ECE estimators, to avoid the sensitive binning choices and reduce the estimation bias.

---

### Author Response · Authors · 2020-11-25
**Response to all reviews**

We thank all reviewers for their thoughtful reviews. We agree with many of the comments raised by the reviewers, and have made a few points for clarification purposes (see our response to AnonReviewer2 for more details.)

---

### Decision · Program_Chairs · 2021-01-07
**Final Decision**

**Decision:**

Reject

**Comment:**

# Paper Summary

This paper considers calibrating the output of a multiclass classifier in such a way that the output probabilities approximately are approximately "correct". They observe that if such a method is able to re-order the logits, then it will change the accuracy of the classifier. Therefore, if they use a calibrator that is constrained to be monotonic in the input logits, then they can train it to optimize any metric they choose, without impacting the accuracy.

They propose minimizing ECE (expected calibration error, which is essentially a binned approximation to the L1 distance between the model's confidence in its top label, and the probability that the top label is correct). Borrowing an idea from local temperature scaling, they also allow their calibrator to see the input features, by also taking the top hidden layer (the layer before the logits) as an input.

Their experiments are, with one glaring exception, comprehensive: they have a good selection of datasets, a reasonable choice of metrics, and they dig pretty deep into what the results mean. Reviewer 2, however, believes that the baselines are far from state-of-the art, and two of the other reviewers (and I) agree.

# Pros

1. Well-organized and well-written (not exceptionally so, but above the bar)
1. Good insight overall. In particular, the observation that imposing a monotonicity constraint enables one to optimize any metric, including ECE, was considered both original and significant by the reviewers
1. Well-thought-out experiments. They were generally praised, aside from the (unfortunately crucial) question of whether the baselines are state-of-the-art

# Cons

1. The paper seems to mainly discuss related work coming from the temperature-scaling "tradition". The reviewers would like to see a more comprehensive discussion of other calibration approaches (Reviewer 2 provided a number of references)
1. There are some misstatements (e.g. that LTS is "state-of-the-art"), and incorrect implications (e.g. that temperature-scaling-like methods are dominant). Reviewer 2 listed several of these, all of which are fairly minor, but more care should be taken
1. The TS and LTS baselines are not state-of-the art. The reviewers were generally impressed with the experiments, but the lack of a strong baseline is a fatal flaw

# Conclusion

This was a paper that initially received mostly-positive reviews, but Reviewer 2 raised several concerns that were not adequately addressed in the author response, causing two other reviewers to lower their scores. Ultimately, three of the four reviewers recommended rejection.

The general consensus is that this is a well-written paper, with good insight, well thought-out experiments (except for the baselines), and that it overall makes a worthwhile contribution. The main issues, all of which were raised by Reviewer 2, are eminently fixable: (i) adding a more thorough discussion of related work, especially work unrelated to temperature scaling, (ii) being more careful to avoid misstatements, or to seem to imply incorrect statements (e.g. that temperature-scaling-like methods are dominant), and (iii) adding a couple of new state-of-the-art baselines to the experiments.